# Annulus fracture underlies radiation-induced sperm dysfunction revealed by multimodal nano-imaging

Nuo Chen[1,2,*], Hongyu Chen[1,*], Jingcheng Xiao[1,2,*], Bo He[3], Difei Zhang[1,2], Xinye Yuan[1,2], Pingping Wen[1,2], Huaidong Jiang[1,2]

**Ionizing radiation impairs male fertility, but the ultrastructural lesions linking radiation to sperm dysfunction remain unclear. The sperm annulus, a ring structure essential for flagellar integrity, is a potential but unresolved target. Here, we integrate synchrotron X-ray ptychography, cryogenic soft X-ray tomography, electron microscopy, and confocal imaging for cross-scale visualization of radiation-induced injury. Ptychography enables quantitative imaging of the annulus at ~40-nm resolution, bridging light and electron microscopy. In irradiated mice, we find annulus fracture, loss of cytomembrane coverage, and disruption at the midpiece-principal piece junction. These defects worsen during maturation, correlating with reduced motility and increased abnormalities. Protein analyses show down-regulation of SEPT12 and AKAP4, indicating compromised annulus and fibrous sheath. Cryogenic tomography confirms ultrastructural deterioration in the axoneme, fibrous sheath, and mitochondria. These findings identify the sperm annulus and fibrous sheath as primary radiation targets, defining a structural basis for radiation-induced male infertility. This multimodal approach offers a generalizable strategy for nanoscale 3D cellular architecture analysis.**

## Introduction

Ionizing radiation represents a pervasive and growing threat to male reproductive health, with significant implications for fertility, offspring well-being, and public health. Its widespread presence in medical, occupational, and environmental settings underscores the urgency of understanding its impact (Fatehi et al, 2018; Kesari et al, 2018; Qin et al, 2021). The male germline, particularly post-meiotic spermatogenic cells and mature spermatozoa, is highly sensitive to radiation-induced damage, leading to transient or permanent infertility, impaired sperm function, and potential genetic risks (Oakberg, 1956; Xu et al, 2008; Georgakopoulos et al, 2024; Yin et al, 2025). Extensive studies on protective agents, such as silymarin and melatonin, have established that oxidative stress, DNA damage, and apoptosis are key mediators of this injury (Khan et al, 2015; Georgakopoulos et al, 2024). These investigations have focused primarily on phenotypic outcomes (e.g., reduced sperm count and motility and increased morphological abnormalities) (Kumar et al, 2013; Zhou et al, 2016) and molecular pathways.

However, a critical gap persists between these functional deficits and a precise, nanoscale understanding of the structural damage inflicted on the spermatozoon. The spermatozoon is a terminally differentiated cell whose unique architecture is inextricably linked to its functions. The compact head safeguards the paternal genome, and the flagellum provides the motility necessary for delivery. Therefore, the structural integrity of the flagellum is fundamental to achieving fertilization. The annulus, a ring-like structure situated at the junction of the middle piece and principal piece of the flagellum, is essential for mechanical stability, compartmentalization, and motility (Kwitny et al, 2010; Chen et al, 2016; Hoque et al, 2024; Whitfield, 2024). The fibrous sheath (FS), a cytoskeletal structure surrounding the principal piece of the flagellum, serves as a rigid scaffold that defines flagellar shape, regulates its bending mechanics, and localizes key signaling proteins essential for motility. Limited by the conventional imaging modalities, direct evidence connecting radiation exposure to specific ultrastructural damage to the annulus or FS has remained elusive (Eddy et al, 2003; Guseva et al, 2024). Confocal microscopy lacks sufficient resolution, whereas electron microscopy often introduces artifacts and fails to capture the three-dimensional context of the native state.

This gap highlights the need for advanced imaging technologies capable of bridging cellular and molecular scales to visualize radiation-induced injury in a near-native state. Ptychography uses coherent monochromatic light to image samples and reconstructs the data with high resolution through an iterative algorithm (Faulkner & Rodenburg, 2004; Rodenburg & Faulkner, 2004), which has demonstrated wide spread use in biology and material science

---

[1]Center for Transformative Science, ShanghaiTech University, Shanghai, China  [2]School of Physical Science and Technology, ShanghaiTech University, Shanghai, China  [3]Medical Radiation Physics, Lund University, Lund, Sweden

Correspondence: jianghd@shanghaitech.edu.cn
*Nuo Chen, Hongyu Chen, and Jingcheng Xiao contributed equally to this work

(Dierolf et al, 2010; Jiang et al, 2010; Holler et al, 2017; Deng et al, 2018; Guo et al, 2022; Michelson et al, 2022) by filling the gap between confocal microscopy and electron microscopy. Here, we deploy a multimodal nanoscale imaging platform integrating synchrotron radiation-based soft X-ray ptychography, cryogenic soft X-ray tomography (cryo-SXT), scanning electron microscopy (SEM), and confocal fluorescence imaging to achieve cross-scale, high-resolution visualization of the sperm ultrastructure. Using this framework, we demonstrate that radiation-induced structural damage in sperm is characterized primarily by annulus fracture and fibrous sheath defects. Using SEM and fluorescence imaging, we preliminarily determined that annulus fracture is associated with disruption of the plasma membrane between the principal piece and midpiece. Western blot analysis further revealed that the fibrous sheath defects result from the down-regulation of the key structural protein AKAP4. Soft X-ray ptychography uniquely provides high-resolution imaging of sperm, and cryo-SXT provides definitive 3D visualization of internal organelles, including the axoneme, mitochondria, and fibrous sheath, in a near-native frozen–hydrated state, confirming ultrastructural deterioration postradiation, which aligns with the observed impairments in sperm function and morphology.

To investigate the nanoscale structural basis of radiation-induced damage to spermatozoa, our study not only identifies the sperm annulus as a primary subcellular target of radiation injury, providing a direct structural explanation for consequent dysfunction, but also establishes a robust and generalizable multimodal imaging strategy. This work provides fundamental mechanistic insight into radiation-induced male reproductive toxicity and introduces a powerful paradigm for investigating nanoscale perturbations in complex biological systems, with implications for developing protective strategies and advancing biomedical research.

# Results

### Gamma irradiation induces reproductive toxicity in mice

To bridge the gap between the known functional deficits and the nanoscale structural integrity of spermatozoa after radiation exposure, we first sought to establish a model of radiation-induced reproductive toxicity. To this end, we confirmed that a single, nonlethal whole-body dose produces canonical reproductive phenotypes at the organ and cellular levels. 8-wk-old male C57BL/6J mice received 5 Gy γ-irradiation at 1 Gy/min (Fig 1A) and were analyzed at 3 d, 1 wk, and 5 wk postexposure (Khan et al, 2015; Yang et al, 2024). Compared with the control treatment, irradiation reduced the size of the testes and significantly decreased the testis-to-body weight ratio (Figs 1B and C and S1A and B). Hematoxylin and eosin staining of the epididymis, the organ responsible for sperm storage and maturation, revealed greater luminal depletion of sperm in irradiated mice than in control mice (Fig 1D). Bright-field imaging revealed reduced sperm output and multiple structural abnormalities (Fig 1E and F), and quantitative analyses

revealed decreased total cauda epididymal sperm counts and a lower proportion of morphologically normal sperm, accompanied by increased frequencies of annulus fracture, hairpin configurations, and headless sperm (Fig 1G–K). In addition to morphological defects, sperm functional competence was also impaired. Compared with control mice, irradiated mice exhibited reductions in multiple motility parameters, including the percentage of total motile sperm, the percentage of progressively motile sperm, the average path velocity (VAP), and the straight-line velocity (VSL), whereas the curvilinear velocity (VCL) remained unchanged (Fig S2B–E). In summary, these results demonstrated that ionizing radiation not only compromised systemic and organ health in mice but also posed a substantial threat to reproductive capacity at the cellular level.

### High-resolution Soft X-ray imaging resolves native sperm architecture

The annulus and adjacent flagellar scaffolds reside below the resolution of conventional light microscopy and can be perturbed by the dehydration and staining required for electron microscopy. We, therefore, implemented label-free soft X-ray ptychography and cryogenic soft X-ray tomography (cryo-SXT) to visualize sperm in near-native states and to calibrate the achievable resolution. In light microscopy images (Fig 1F), the principal piece (PP) and middle piece (MP) of spermatozoa can be clearly distinguished, and the annulus resides at the junction of these two regions. Compared with the optical image of the spermatozoon (Fig 2A), the soft X-ray ptychography reconstruction result (Fig 2B) showed higher resolution and contrast. Benefit from the advantages of ptychography, the annulus and junction between the head and middle piece, which can easily undergo structural changes upon radiation exposure, were characterized at high resolution (Fig 2C and D). The two structures shown here were normal and intact because the sample came from a healthy mouse with no exposure to ionizing radiation.

To quantitatively assess the resolution of the ptychographic phase projection (Fig S3A), we first calculated the pixel size of the reconstructed images using the formula $d = \lambda/2\sin(\theta)$, where $\lambda$ is the wavelength and $\theta$ is the diffraction angle. In our experiment, the pixel size was 9.99 nm. The power spectral density (PSD) curve of the 2D diffraction patterns (Fig S3B) showed that the diffraction signal extended to a spatial frequency of 72 $\mu m^{-1}$, corresponding to a theoretical real-space resolution of 13.9 nm. In addition to this, the Fourier Ring Correlation resolution of this experiment reached 41.7 nm (Fig S3C).

Soft X-rays penetrate biological materials better than electrons do, permitting the imaging of specimens as thick as 10 $\mu m$. Unlike electron microscopy, this technique does not require preimaging sectioning of eukaryotic cells with an ultramicrotome. Moreover, since image contrast arises directly from differential X-ray absorption, dehydrating or staining the specimens is unnecessary. In the absence of heavy metal stains, compositional information can be detected with no interference. Consequently, cryogenic soft X-ray yields high-resolution images of cellular structures preserved in a near-native state (Carzaniga et al, 2014; Duke et al, 2014). To more accurately characterize the ultrastructure of the

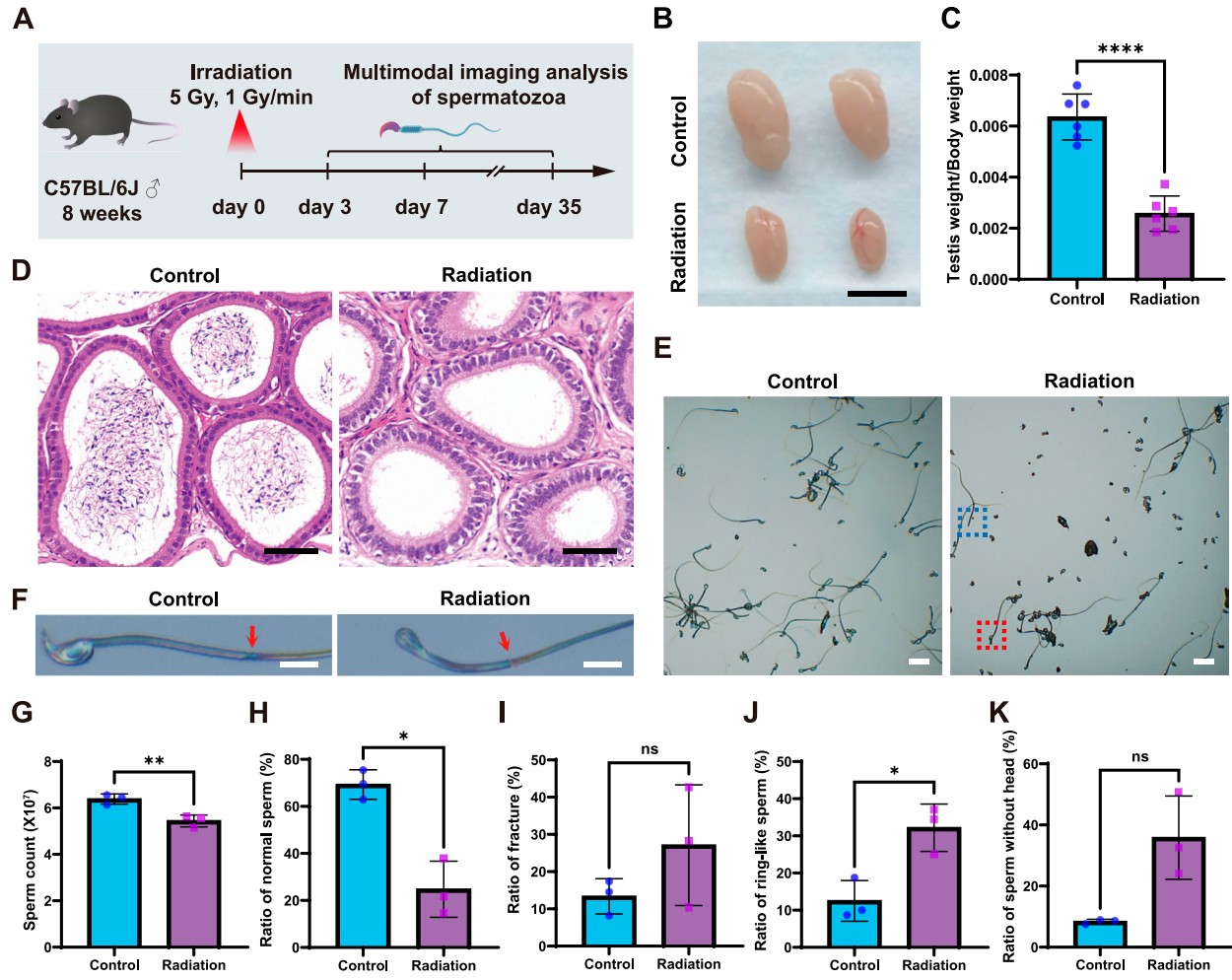

**Figure 1. Phenotypic characterization of testes and spermatozoa in the irradiated mouse model.**
**(A)** Schematic overview of the irradiation regimen and sampling timeline for 8-wk-old male C57BL/6J mice (a single dose of 5 Gy at 1 Gy/min on day 0; multimodal analyses on days 3, 7, and 35). **(B)** Pictures of representative testes from control and irradiated mice. **(C)** Testis-to-body weight ratio in control and irradiated mice. Scale bar, 0.5 mm. **(D)** H&E staining of cauda epididymal sections from control and irradiated mice. Scale bar, 600 $\mu$m. **(E)** Bright-field images of cauda epididymal sperm from control and irradiated mice. Scale bar, 25 $\mu$m. The representative headless sperm and hairpin structure were marked by blue and red rectangles, respectively. **(F)** Magnified bright-field view of the sperm annulus and fracture (red arrows). Scale bar, 5 $\mu$m. **(G, H, I, J, K)** Quantification of cauda epididymal sperm counts (G), proportion of morphologically normal sperm (H), proportion of sperm with annulus fracture (I), proportion of sperm with hairpin structure (J), and proportion of headless sperm (K) in control versus irradiated mice. The data are presented as the mean ± SEM. Statistical analysis by $t$ test: ns, not significant; *$P$ < 0.05; **$P$ < 0.01; ****$P$ < 0.0001. Scale bars, as indicated.

spermatozoa, we used cryogenic soft X-ray tomography. The ax-oneme, fibrous sheath, and mitochondria were segmented by Amira and clearly distinguishable in 3D with a 16.9-nm spatial resolution (Fig 2E, Video 1). In summary, soft X-ray ptychography and cryo-SXT can deliver label-free, near-native visualization of the flagellar ultrastructure and provide quantitative resolution benchmarks.

### Radiation induces structural fractures in the sperm annulus

To investigate whether ionizing radiation can induce structural damage to the sperm annulus, we used soft X-ray ptychography in combination with SEM. Ptychography was used to image the principal piece and middle piece of both irradiated and normal sperm (Fig 3A–D). The ptychographic images

enabled the visualization of structural alterations that are difficult to resolve with conventional optical microscopy, es-pecially the extremely fine structure of the sperm annulus (Fig S4A–F).

To quantify radiation-induced damage to the annulus, we measured the ratio of annulus diameter to midpiece diameter in the control and irradiated groups (Fig 3E). In the control group, this ratio was greater than 1, indicating that the annulus forms a ring-like junction structure at the boundary between the midpiece and principal piece and is expected to appear as an intact circum-ferential constriction. This structure can also be observed in the 3D reconstruction result (Video 2), the 3D resolution of which was analyzed by line scan (Fig S5A–D). By contrast, in the irradiated group, the ratio was less than 1, indicating annulus fracture and exposure of the axoneme.

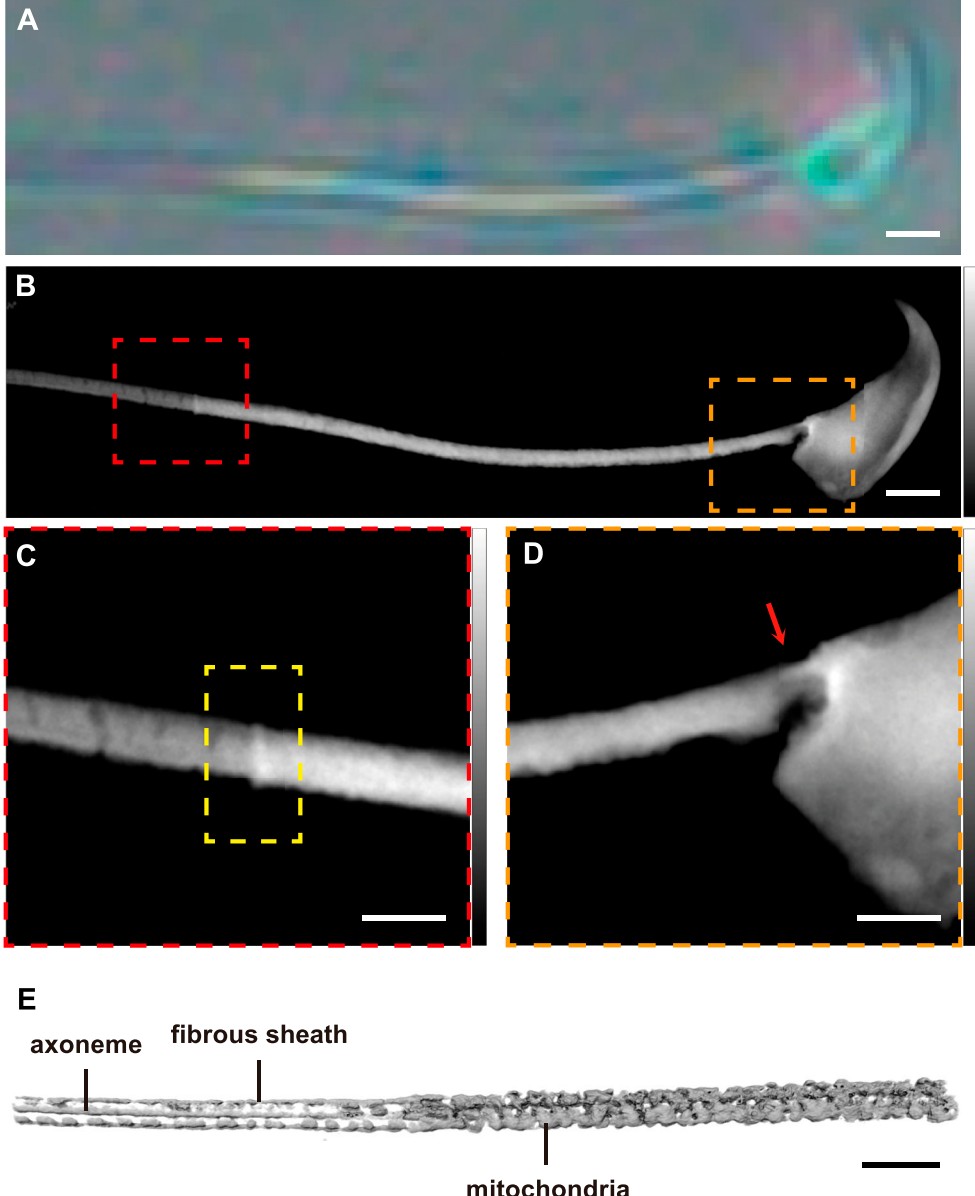

**Figure 2.   Correlative optical and soft X-ray imaging of a normal mouse spermatozoon.**
**(A)** Bright-field optical micrograph of a single sperm cell. **(B)** Ptychographic soft X-ray phase map of the same cell. **(B, C, D)** Enlarged views of the boxed regions in panel (B); in (C), the annulus region is outlined by a yellow dashed rectangle; and in (D), the head–midpiece junction is indicated by a red arrow. **(E)** Cryogenic soft X-ray transmission computed tomography reconstruction of a sperm flagellum, with the axoneme, fibrous sheath, and mitochondrial sheath discernible. Scale bars: (A, B) 2 $\mu m$; (C, E) 1 $\mu m$.

Furthermore, SEM analysis (Fig 3F and G) yielded consistent results. We also measured the fracture width of the annulus in irradiated and normal sperm (Fig 3H). The extent of annulus disruption was clearly much greater in the irradiated group than in the control group. Together, these data demonstrate that ionizing radiation directly induces structural fractures in the sperm annulus, thereby achieving the initial objective of confirming radiation as a cause of annulus damage.

We assessed the motility parameters, such as total motility, progressive motility, VAP, VSL, and VCL of spermatozoa from control and irradiated mice (Fig S2A–E) and found that the radiation-induced fracture in the annulus may be related to overall sperm motility parameters, indicating severe compromise of the reproductive and fertilization capacity of these mice. This finding links the observed structural damage of the annulus to a critical functional deficit, suggesting that the radiation-induced fracture impairs sperm motility.

### Radiation induces time-dependent structural deterioration in the sperm midpiece and principal piece

Mature cauda sperm collected 5 wk after exposure were derived from spermatogonial cohorts irradiated early; we investigated whether the annulus and adjacent compartments deteriorated as the irradiated cohorts transitioned from spermatogenesis through epididymal maturation. In spermatozoa from irradiated mice, we

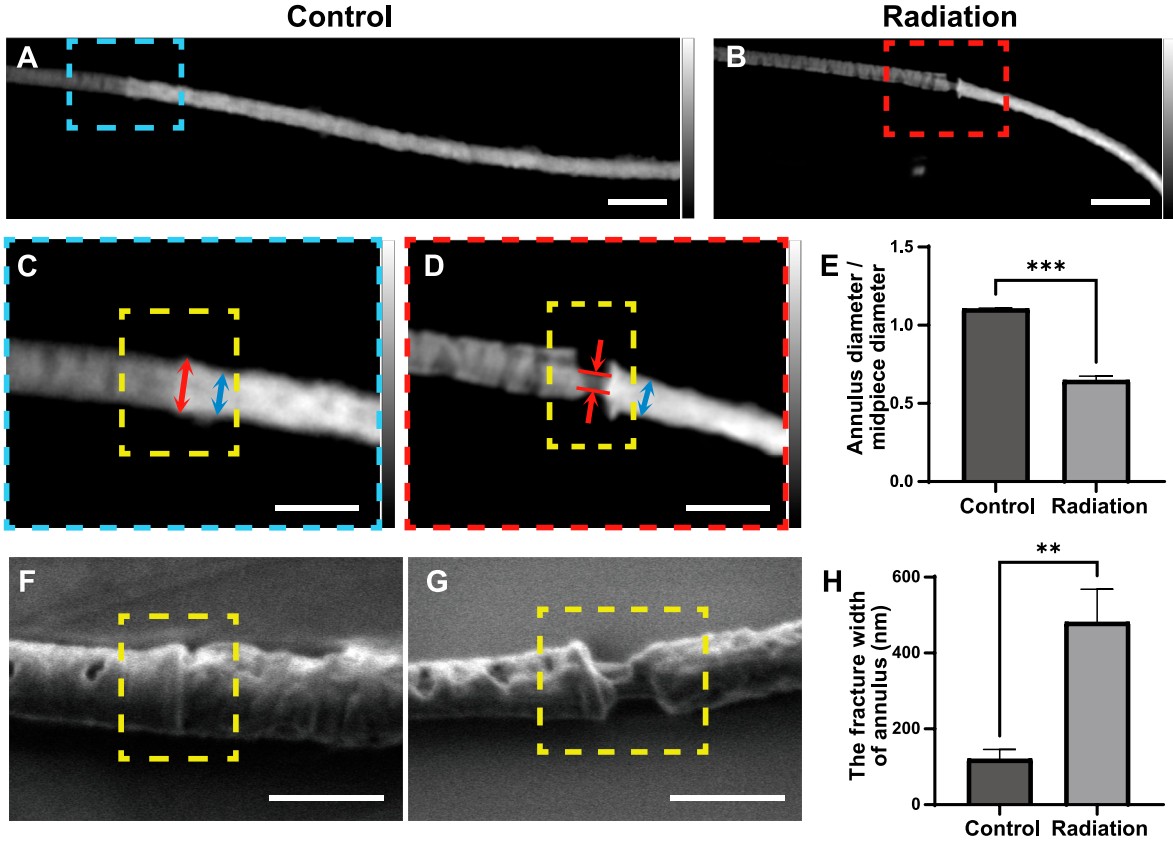

**Figure 3.  Ptychography and SEM revealed annulus abnormalities in irradiated mouse sperm.**
**(A)** 2D ptychographic phase map of a spermatozoon from a control mouse. Scale bar, 2 μm. **(B)** 2D ptychographic phase map of a spermatozoon from a mouse exposed to radiation. Scale bar, 2 μm. **(A, B, C, D)** Enlarged views of the boxed regions in (A, B); blue arrows indicate the midpiece diameter; and red arrows indicate the annulus diameter. Scale bar, 1 μm. **(E)** Comparison of the annulus-to-midpiece diameter ratio between the control and irradiated groups. **(F, G)** Scanning electron microscopy images of the annulus region from control (unexposed; (F)) and irradiated (5 wk postexposure; (G)) mice; the annulus is highlighted by yellow dashed rectangles. Scale bar, 1 μm. **(H)** Quantification of the annular fracture width (nm) in sperm from irradiated versus control mice. Bar graph data are presented as the mean ± SEM. Statistical analysis by *t* test; significance is indicated as **\*\*P* < 0.01; \*\*\*P* < 0.001.

observed not only fractures in the annulus but also different degrees of structural disruption in the middle piece (MP) and principal piece (PP), which are adjacent to the annulus. Sperm collected 3 d after irradiation exhibited relatively intact MP and PP regions, and the segments adjacent to the annulus remained closely wrapped around the axoneme (Fig 4A and D). In contrast, sperm collected 1 wk after irradiation showed some notches, i.e., fissures with sides converging at a distinct angle, in the PP region (Fig 4B and E), along with splitting in the region adjacent to the annulus, although the MP did not display notable changes. At ~5 wk, which corresponds to a late stage of sperm maturation in mice, compared with the earlier time points, the PP adjacent to the annulus also exhibited splitting and visible gaps, i.e., fissures bounded by nearly parallel sides. The MP was no longer tightly wrapped around the axoneme and instead opened outward in a petal-like manner (Fig 4C and F). This damage adjacent to the annulus was temporally progressive, culminating in pronounced midpiece and principal piece disorganization in late-maturing sperm. The structural injury originated at and near the annulus and accumulated over time, which is consistent with the

progressive failure of junctional scaffolds and coverings as irradiated cohorts mature. These temporal observations confirm that radiation-induced damage to the annulus initiates a progressive structural deterioration in adjacent sperm compartments throughout maturation, characterizing the evolution of this injury over time.

## Principal piece microlesions increase over time after irradiation

To link structural injury to potential biomechanical and signaling consequences, we quantified the abundance and geometry of principal piece microlesions. SEM images (Fig 5A and B) revealed that the damage to the PP region of spermatozoa collected 5 wk after irradiation was greater and deeper than that of normal spermatozoa. To better illustrate the structural difference, we used X-ray ptychography by using the contrast difference to analyze the depth information. The contrast differences in the ptychography reconstruction results revealed that the notches observed in the PP region of control sperm

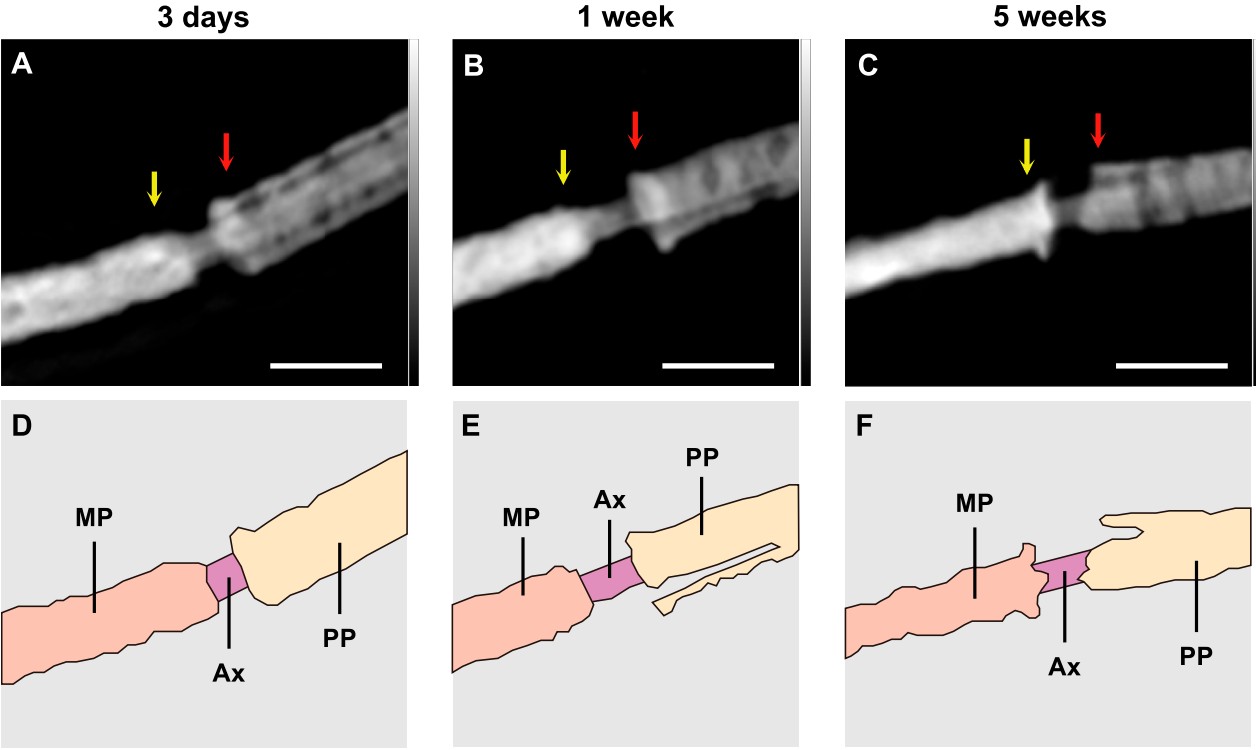

**Figure 4.   Time-dependent changes at the annulus (midpiece–principal piece junction) in mouse sperm after irradiation.**
**(A, B, C)** 2D ptychographic phase contrast images of the annulus region in sperm collected 3 d (A), 1 wk (B), and 5 wk (C) after irradiation; yellow arrows indicate the midpiece, and red arrows indicate the principal piece. Scale bar, 1 $\mu$m. **(A, B, C, D, E, F)** Corresponding schematic diagrams for (A, B, C), illustrating the midpiece, axoneme, annulus, and principal piece.

exhibited only minor contrast variation (Fig 5C and D). In contrast, spermatozoa collected 5 wk after irradiation displayed notches with pronounced contrast changes, which may indicate more severe damage to both the plasma membrane and the fibrous sheath. To this extent, the results of the two imaging methods converge.

To better understand whether different time periods after irradiation are associated with the occurrence of notches and gaps in the PP region, we quantified these features in ptychographic images of spermatozoa from mice exposed to the same radiation dose but sampled at different time points. Specifically, we counted the number of notches and gaps per micrometer along the axoneme from the annulus toward the PP region. The number of notches and gaps increased progressively with later sampling times, suggesting that radiation caused more severe membrane and fibrous sheath damage in spermatozoa that matured later (Fig 5E). In addition, the opening angles of the notches in the PP region also varied across different sampling time points (Fig 5F). This structural change may be related to the observed impairment in sperm motility. The time-dependent surge in microlesion number and altered geometry is consistent with progressive damage to the plasma membrane and fibrous sheath, with expected effects on bending stiffness, beat symmetry, and compartmentalized signaling along the principal piece. In summary, radiation drives quantitative and geometric remodeling of principal piece microlesions, indicating increased scaffold and membrane failure.

### Annulus fracture involves protein down-regulation and localized membrane damage

To relate ultrastructural defects to underlying macromolecular changes, we assessed key structural proteins and membrane coverage at the annulus. Western blots revealed notable down-regulation of AKAP4 (fibrous sheath scaffold) (Turner et al, 2001) and SEPT12 (annulus-associated septin) (Shen et al, 2017) in irradiated epididymal lysates (Fig 6A and B), which substantiated fibrous sheath and annulus deterioration. Although we observed annulus fractures in ptychography, the structure appeared to remain intact under SEM (Fig 3). To further investigate the cause of fracture at the annulus region, we stained the sperm with Nile Red and found that the disruption of sperm lipid components precisely localized to the site of annular fracture (Fig 6C). A general plasma membrane stain (CellMask) and a phospholipid-specific probe further confirmed that the annular fracture was associated with localized plasma membrane damage (Fig 6D). These results corroborate that the annulus and fibrous sheath compromise and support a model in which annular fracture is accompanied by local membrane depletion.

## Discussion

The integrity of the sperm flagellum, particularly its specialized cytoskeletal structures such as the annulus and fibrous sheath, is

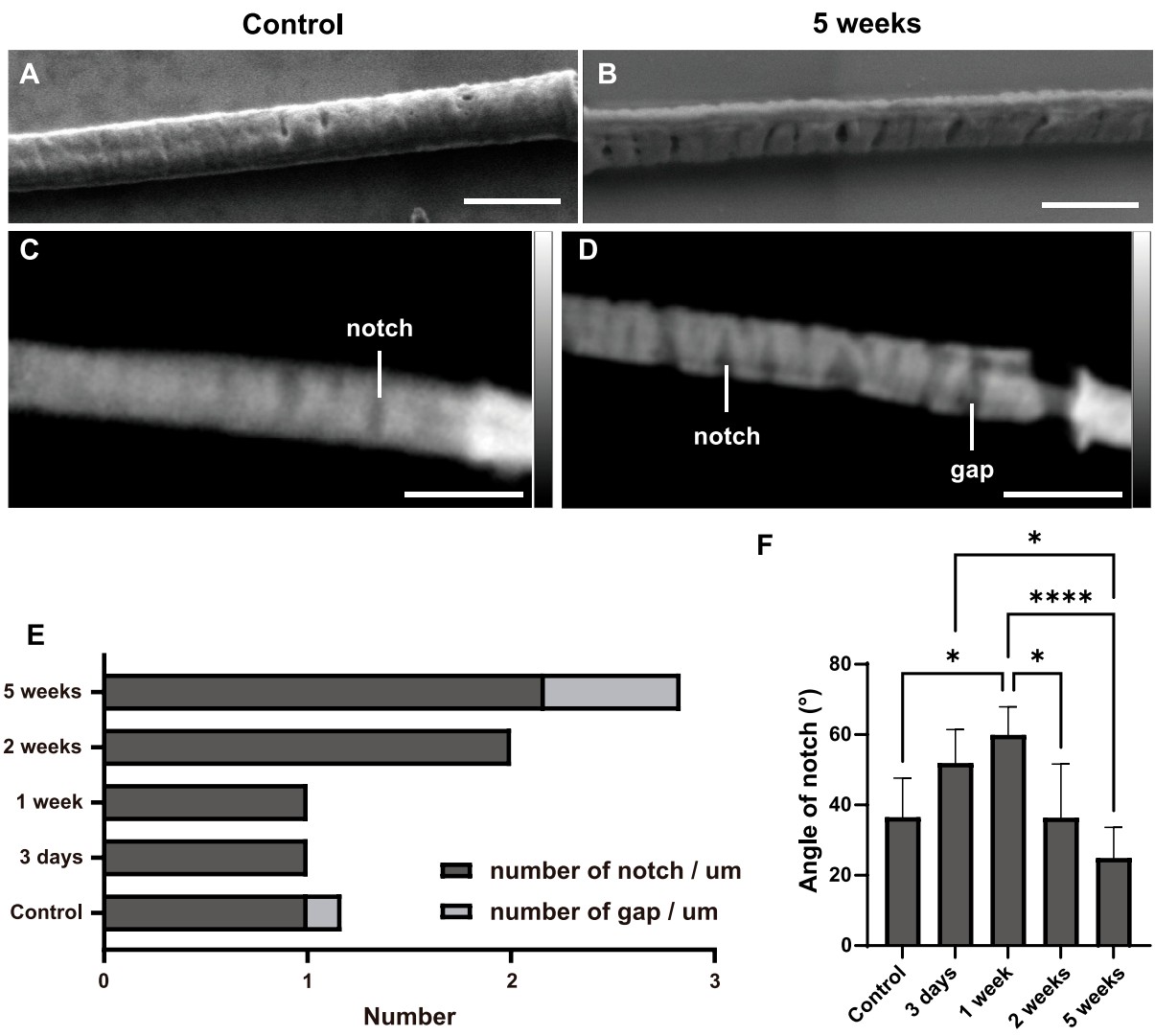

**Figure 5. Notch and gap defects in the principal piece of mouse sperm after irradiation.**
**(A, B)** Scanning electron microscopy images of the principal piece from control mice (A) and mice 5 wk after irradiation (B). Scale bar, 1 $\mu$m. **(C, D)** 2D ptychographic phase maps of the principal piece from control sperm (C) and sperm 5 wk after irradiation (D); notches and gaps are indicated as labeled. Scale bar, 1 $\mu$m. **(E)** Quantification of the number of notches and gaps per micrometer along the principal piece in sperm collected at the indicated time points after irradiation. **(F)** Notch angle measurements over time after irradiation. The data in the bar graphs are presented as the mean ± SEM. Statistical analysis by one-way ANOVA; *$P < 0.05$; ****$P < 0.0001$.

fundamental to its motility and function. However, a precise, nanoscale understanding of how environmental insults like ionizing radiation compromise this structural integrity has remained elusive, creating a critical gap between observed functional deficits and their underlying physical causes. This study aimed to bridge this gap by deploying a multimodal nanoscale imaging platform to directly visualize and characterize the structural damage inflicted by radiation on spermatozoa in a near-native state. Here, we identify the sperm annulus as an important structural target of ionizing radiation and delineate how damage at this junction propagates to adjacent flagellar compartments and impairs function. Using a single sub-lethal whole-body dose, we recapitulated canonical reproductive toxicity at the organ and cellular scales—reduced testis mass, depleted cauda epididymal sperm,

and diminished motility—and then resolved nanoscale injury with a cross-scale imaging pipeline. It should be noted that 5 Gy is considered to be a high-dose experimental/sub-lethal exposure condition, mainly relevant to accidental exposure scenarios, planned radiation exposure contexts, or mechanistic studies of radiation-induced reproductive toxicity (Khan et al, 2015). Soft X-ray ptychography and cryo-SXT revealed that irradiation converts an intact, widened annulus into a narrowed, fractured barrier, reverses the annulus–midpiece diameter relationship, and is accompanied by principal piece microlesions that increase in number and change in geometry over time. These structural changes co-occur with the down-regulation of AKAP4 and SEPT12 expression and reduced lipid signaling at the annulus, linking flagellar scaffold failure and local membrane loss to

progressive motility impairment. In contrast, transmission electron microscopy did not reveal obvious or significant ultrastructural alterations in midpiece mitochondria under our experimental conditions (Fig S6A–D), suggesting that mitochondrial damage is not the predominant ultrastructural feature in this model. In addition, available evidence indicates that spermatogenesis can be at least partially re-established after irradiation, depending on radiation dose, dose rate, and the post-irradiation interval (Bae et al, 2021). Consistent with this interpretation, histological analysis of the testes showed a loss of elongated spermatids in the seminiferous tubules after irradiation (Fig S7A–D), and epididymal H&E staining revealed marked luminal sperm depletion, suggesting that the abnormalities observed in mature cauda epididymal sperm likely originated, at least in part, from defects arising during spermatogenesis in the testis. However, this possibility needs to be further investigated under our experimental conditions.

The results support a mechanistic model in which the annulus and the fibrous sheath form a coupled mechanical and diffusional interface that is preferentially compromised by radiation (Kwitny et al, 2010). Annulus fracture likely weakens the circumferential constraint at the midpiece-principal piece junction and disrupts the septin-based barrier, allowing local membrane depletion and destabilization of the overlying fibrous sheath. The progressive emergence of notches and gaps with evolving opening angles along the principal piece, together with outward "petal-like" midpiece separation from the axoneme at later time points, is consistent with the gradual loss of structural integrity during spermatogenesis and epididymal maturation of irradiated cohorts. Functionally, the observed reduction in VAP and VSL with preserved VCL indicates a decline in path coherence rather than instantaneous speed, as expected, when longitudinal scaffolds and regional membrane organization that tune bend stiffness and waveform propagation are disrupted.

Beyond establishing causative structural motifs of injury, our work advances methodology by bridging a persistent resolution gap. Label-free ptychography provided high-contrast, quantitative phase maps with an empirical ~40-nm Fourier Ring Correlation resolution, enabling measurement of annular and midpiece diameters in near-native states. Ptychography delivered volumetric context and axis-dependent resolution in the range of 65–94 nm (Giewekemeyer et al, 2010). In concert with SEM and fluorescence imaging, this correlative pipeline allowed us to connect nanoscale architecture, membrane coverage, and protein expression within the same experimental framework, thereby strengthening inferences about the spatial origin and progression of flagellar damage.

Based on the loss of membrane integrity at the annulus region, we propose that this defect may result from dysregulation of membrane glycerophospholipid metabolism. Radiation is known to perturb membrane lipids via oxidative reactions and metabolic remodeling, potentially altering phospholipid composition, asymmetry, and biophysical stability (Wolters et al, 1987; Mishra, 2002). However, our current data cannot distinguish lipid loss from lipid reorganization or identify which lipid classes are altered. Future work should therefore apply lipidomics or metabololipidomics on purified sperm to test whether glycerophospholipid pathways are selectively disrupted and whether such changes correlate with annulus fracture severity.

In summary, our multimodal nanoscale imaging reveals that radiation induces annulus fracture and fibrous sheath damage in mouse sperm, directly linking these ultrastructural injuries to impaired motility. Together, our findings position the annulus–fibrous sheath–membrane axis as the structural and molecular fulcrum of radiation-induced sperm dysfunction and provide a generalizable framework for evaluating environmental injury in highly differentiated cells.

# Materials and Methods

### Animals and gamma irradiation

Eight-wk-old male C57BL/6J mice were purchased from Shanghai Yishang Biotechnology. All the mice were housed in a specific pathogen-free barrier facility under a controlled environment with a 12-h light/12-h dark cycle, a temperature of 22 ± 2°C, and a relative humidity of 50–60%. Food and sterile water were provided ad libitum throughout the experiment. After a 7-d adaptive feeding period to allow the mice to acclimatize to the housing conditions, a total of 24 mice were randomly divided into two groups with 12 mice per group. Mice in the control group were maintained under normal feeding conditions without any irradiation, and those in the irradiated group received total body irradiation with γ-rays using a $^{137}$Cs radioactive source (Gammacell 40 Exactor, ON K2K OE4; Best Theratronics). The irradiation was administered at a dosage rate of 1 Gy/minute, delivering a total absorbed dose of 5 Gy. After irradiation, mice in both groups were continuously housed under the aforementioned standard conditions for different observation periods (3 d, 1 wk, or 5 wk) before sample collection. All animal handling and experimental procedures were conducted in compliance with the Guidelines for the Care and Use of Laboratory Animals and were approved by the Animal Care and Use Committee (ACUC) of ShanghaiTech University (Approval No. 20231215001).

### Histological analysis

Immediately after euthanasia, the epididymides were carefully dissected from the mice, fixed in 4% PFA at room temperature for

**Figure 6. Western blot and confocal imaging revealed protein loss and annular defects in irradiated mouse sperm.**
**(A)** Immunoblot analysis of AKAP4 and SEPT12 in control and irradiated epididymides; GAPDH served as the loading control. **(B)** Densitometric quantification of AKAP4 and SEPT12 expression in control and irradiated epididymides (n = 3); data are shown as the mean ± SEM. Statistical analysis by *t* test: ***P < 0.001; ****P < 0.0001. **(C, D)** Confocal fluorescence images of cauda epididymal sperm from control mice and mice 5 wk after irradiation; Hoechst was used to label nuclei (blue). BF, bright field. Red arrows indicate the normal annulus in control sperm; blue arrows indicate the annulus structure in irradiated sperm. A magnified view of the dashed box in C is shown at the upper right; the dashed contour delineates a missing head. Scale bar, 5 μm.

24 h to preserve histological integrity, and then processed via gradient ethanol dehydration and embedded in paraffin. Afterward, the tissue blocks were cut into 4-$\mu$m sections and mounted on glass slides. After being baked at 60°C for 2 h to enhance adhesion, the slides underwent routine deparaffinization and hematoxylin/eosin (H&E) staining. Images were captured using a digital slide scanner (Pannoramic ScanII; 3D HISTECH).

## Sperm morphological classification criteria and quantification method

Sperm were classified based on bright-field morphology. Morphologically normal sperm are sperm with an intact head, a continuous connection between the midpiece and principal piece flagellum, and no obvious folding or bending of the flagellum into a nearly parallel configuration. Annulus-fractured sperm are sperm showing an obvious a clear discontinuity, narrowing, or break-like defect at the junction between the midpiece and principal piece. Hairpin-/ring-like bent sperm are sperm in which the flagellum was folded at the annulus region to an angle close to 180°, such that the midpiece and principal piece became nearly parallel. Headless sperm are sperm lacking attached heads.

For the quantification of sperm abnormalities, bright-field images of cauda epididymal sperm were collected from three mice per group. For each group, we analyzed at least 18 microscopic fields in total. The percentage of the normal and three abnormal categories was then calculated as: number of normal or abnormal sperm/total number of sperm counted × 100%.

## Sperm motility assessment

To evaluate sperm quality, the cauda epididymides from each mouse in both groups were carefully dissected and minced with sterile scissors in 1 ml of pre-warmed (37°C) PBS to release spermatozoa for 30 min. After incubation, 10 $\mu$l of the supernatant containing motile sperm was used for the analysis of the sperm concentration and multiple motility parameters, including progressive motility, average path velocity (VAP), curvilinear velocity (VCL), and straight-line velocity (VSL), by a computer-assisted sperm analysis system (CEROS II; Hamilton Thorne, Inc.) after the manufacturer's recommended settings. All measurements were performed in biological triplicates to ensure data reliability.

## Dehydrated and cryogenic sperm sample preparation

Freshly excised caudal epididymides of mice were minced and placed in HTF solution at 37°C for 10 min to allow the release of sperm. Afterward, the mixture was centrifuged at 100$g$ for 2 min, and the supernatant containing the sperm was collected, after which the epididymal debris at the bottom was discarded. The supernatant was subsequently centrifuged at 500$g$ for 3 min, after which it was discarded to obtain the sperm sediment at the bottom. Then, the sperm were prefixed with 4% PFA for 30 min. The prefixed sperm were washed three times with ddH$_2$O. The sperm were subsequently gradually dehydrated in 30%, 50%, 70%, 80%, 90%, and then 100% ethanol for 15 min per concentration (Chen

et al, 2014). Finally, the sperm were dripped onto a silicon nitride window and then left to dry in a drying cabinet. Since the X-ray absorption of a 30-nm-thick Si$_3$N$_4$ membrane is relatively low, the effect of the substrate on the imaging results could be reduced by attaching cells to this kind of membrane.

To prepare cryogenic X-ray tomography samples, X-PIVOT grids (XP-FF-MO100M) were glow-discharged for 30 s. Two groups of sperm samples in PBS were applied to each grid in the Vitrobot Mark IV (Thermo Fisher Scientific), blotted for 1 s at 4°C at 100% humidity, and plunge-frozen into liquid ethane. The frozen grids were transferred to liquid nitrogen and stored for data acquisition.

## Ptychography imaging

Ptychography experiments were carried out on BL08U1A at the Shanghai Synchrotron Radiation Facility (SSRF) and on beamline 101D-1 at the Canadian Light Source. The data of samples under the effect of radiation were collected on BL08U1A at the SSRF. During the SSRF ptychography experiments, the photon energy of the incident X-ray was set to 520 eV for the ptychography imaging. In the ptychography experiment, the samples were raster scanned point by point with a 2 $\mu$m-diameter probe at a step size of 400 nm, which corresponds to 80% overlap between adjacent scanning areas. The mixed-states ePIE algorithm was used to conduct the ptychographic reconstruction (Maiden & Rodenburg, 2009; Thibault & Menzel, 2013). The diffraction patterns at each location of the samples were collected with the detector in the far field.

The data of the control sample for 3D reconstruction, a total of 50 angles of projections from 48° to –75°, were collected on beamline 101D-1 at the Canadian Light Source. Every 2D ptychography projection of these 3D data used a 2.5-$\mu$m-diameter probe at a step size of 500 nm, which corresponds to 80% overlap between adjacent scanning areas. The same algorithm was used to conduct the ptychographic reconstruction.

## Cryogenic soft X-ray tomography

Cryogenic soft X-ray tomography was performed at beamline BL07W of the National Synchrotron Radiation Laboratory. A pre-prepared molybdenum grid with cryogenic samples was inserted into a sample holder in liquid nitrogen. Then, the sample holder with the grid was transferred to the chamber of a transmission soft X-ray microscope. The fixed cells were observed via the transmission soft X-ray microscope at an X-ray energy of 535 eV. To construct the 3D volume, the cells were rotated from –65° to +65°, and a continuous series of 131 projected images was captured at 1° intervals with an exposure time of 1 s.

## 3D reconstruction method

All projections were first aligned to the 0° reference projection using a cross-correlation method. After background removal, normalization was performed using the common-line method. CT reconstruction was then carried out with the real-space iterative reconstruction algorithm (RESIRE) (Pham et al, 2023). After the

initial reconstruction, all projections were further registered to the corresponding back-projection images via feature-point matching to achieve subpixel-level alignment correction. The final reconstruction was subsequently obtained using RESIRE. The Amira software package was used to display and segment the reconstructed sperm.

### Scanning electron microscope

Sperm were collected from the cauda epididymis and prefixed with 4% PFA for 30 min. The prefixed sperm were subsequently washed three times with ddH$_2$O. The sperm were then gradually dehydrated in 30%, 50%, 70%, 80%, 90%, and then 100% ethanol for 10 min per concentration. Finally, the sperm were dripped onto an Si$_3$N$_4$ window and then left to dry in a drying cabinet. Si$_3$N$_4$ windows were used because they cause less contamination during manufacturing, and their surfaces do not contain contaminants that affect imaging. Moreover, they are also convenient for subsequent X-ray imaging experiments on samples. After natural drying, SEM was performed using a JEOL JSM-7800F Prime operated in high-vacuum mode with a secondary electron detector at an accelerating voltage of 1 kV and working distance of 9.9 mm.

### Confocal fluorescence imaging

Sperm cells were isolated from the cauda epididymis of control and irradiated mice and released into pre-warmed PBS. After gentle dispersion, the sperm suspension was washed with PBS by centrifugation to remove tissue debris. For fluorescence staining, sperm cells were incubated in PBS containing Nile Red (1 $\mu$M, 60530ES03; Yeasen), CellMask (1:1,000, C10046; Thermo Fisher Scientific), phospholipid dye (1:1,000, H34351; Thermo Fisher Scientific), or Hoechst33342 (1:100, C0031; Solarbio) for 30 min at 37°C in the dark. After staining, cells were washed twice with PBS by centrifugation to remove excess unbound dye, mounted on confocal dishes with antifade mounting medium (P0126; Beyotime), and imaged immediately.

Confocal fluorescence and bright-field images were acquired using an Olympus FV3000 laser scanning confocal microscope (equipped with 405-, 488-, 561-, 594-, and 640-nm lasers, two standard high-resolution photomultiplier tube PMT detectors, and two high-sensitivity cooled GaAsP detectors). Images were obtained with a 100× (NA: 1.4, oil immersion) objective. Hoechst33342, Nile Red, phospholipid dye, and CellMask were excited using the 405-, 561-, 594-, and 640-nm laser lines, respectively. Emission signals were collected using spectral detection windows matched to each fluorophore, and channels were acquired sequentially to reduce fluorescence bleed-through.

For each experiment, confocal acquisition parameters, including laser power, detector gain, offset, pinhole size, scan speed, zoom factor, and image resolution, were kept constant between control and irradiated samples. Image processing was performed using Olympus FV31S-SW software and ImageJ. For representative images, brightness and contrast were adjusted linearly and applied equally to all images within the same comparison group.

### Western blot analysis

The caudal epididymis from control and irradiated mice was lysed in RIPA buffer (89900; Thermo Fisher Scientific), and the proteins were then separated by 10% SDS–PAGE and blotted onto polyvinylidene fluoride (PVDF) membranes (88520; Thermo Fisher Scientific). After blocking in 5% skim milk at room temperature for 1 h, the membranes were incubated with an anti-Akap4 (1:1,000, A14813; ABclonal), anti-Sept12 (1:2,000, PH19019S; Abmart), or anti-Gapdh (1:10,000, ET1601-4; Huabio) antibody at 4°C overnight and with an HRP-linked antibody (#7074; CST) at room temperature for 1 h. The blots were imaged using ChemiScope 6100 (Clinx).

### Statistical analysis

The results are expressed as the mean ± SE of the mean. All the statistical analyses were performed using the statistical software GraphPad Prism (version 10.1.1; GraphPad Software). $P < 0.05$ was regarded as statistically significant. Differences between the controls and other samples were assessed with $t$ tests. One-way ANOVA was used to determine the difference in more than two groups.

## Data Availability

The data that support the findings of this study and details of the reconstruction algorithm are available from the corresponding authors on reasonable request.

## Supplementary Information

## Acknowledgements

We acknowledge the staff at the CLS 101D-1 beamline and the SSRF beamline 08U1A for their assistance with data acquisition. The authors thank the support from the Analytical Instrumentation Center (Contract No. SPST-AIC10112914), School of Physical Science and Technology, ShanghaiTech University. This work was supported by the National Key R&D Program of China (2022YFA1603703), the National Natural Science Foundation of China (12335020), and Basic Research Program Based on Major Scientific Infrastructures (JZHKYPT-2021-05).

### Author Contributions

N Chen: conceptualization, data curation, formal analysis, investigation, visualization, and writing—original draft.
H Chen: conceptualization, data curation, formal analysis, validation, investigation, and writing—original draft.
J Xiao: data curation, software, formal analysis, methodology, and writing—review and editing.
B He: data curation.
D Zhang: data curation.

X Yuan: data curation.

P Wen: data curation.

H Jiang: conceptualization, resources, supervision, funding acquisition, project administration, and writing—review and editing.

## Conflict of Interest Statement

The authors declare that they have no conflict of interest.

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
