## [Reviewer comments · Life Science Alliance]

Annulus fracture underlies radiation-induced sperm dysfunction revealed by multimodal nano-imaging

Nuo Chen, Hongyu Chen, Jingcheng Xiao, Bo He, Difei Zhang, Xinye Yuan, Pingping Wen, and Huaidong Jiang
DOI: <https://doi.org/10.26508/lsa.202603693>

Corresponding author(s): Huaidong Jiang, ShanghaiTech University

Review Timeline:

Submission Date:	2026-03-12
Editorial Decision:	2026-04-20
Revision Received:	2026-04-26
Editorial Decision:	2026-05-12
Revision Received:	2026-05-14
Accepted:	2026-05-14

Scientific Editor: Tim Fessenden

Transaction Report:

April 20, 2026

Re: Life Science Alliance manuscript #LSA-2026-03693-T

Prof. Huaidong Jiang
ShanghaiTech University
393 Middle Huaxia Road, Pudong New Area
Shanghai 201210
China

Dear Dr. Jiang,

Thank you for submitting your manuscript entitled "Annulus fracture underlies radiation-induced sperm dysfunction revealed by multimodal nano-imaging" to Life Science Alliance. The manuscript was assessed by expert reviewers, whose comments are appended to this letter.

As you will see, both reviewers expressed support for these findings on sperm morphology made possible by soft X-ray ptychography and Cryo-SXT. We invite you to submit a revised manuscript addressing all concerns by the reviewers. We particularly encourage you to expand the sperm morphology analysis shown in Fig 1, per comment 2 by Reviewer 1, and provide both intensity and phase maps per point 1 by Reviewer 2.

I would be happy to discuss the revision in more detail via email or phone/videoconferencing. Please let me know which option you prefer, if any.

While you are revising your manuscript, please also attend to the below editorial points to help expedite the publication of your manuscript. Please direct any editorial questions to the journal office. When submitting the revision, please include a letter addressing the reviewers' comments point by point.

Thank you for this interesting contribution to Life Science Alliance. We hope that the comments below will prove constructive as your work progresses, and we are looking forward to receiving your revised manuscript.

Sincerely,

B. MANUSCRIPT ORGANIZATION AND FORMATTING:

Reviewer #1 (Comments to the Authors (Required)):

The authors reported in this manuscript analyses of sperm ultrastructure using multiple imaging methods including confocal microscopy, synchrotron radiation based soft X-ray ptychography, Cryo-SXT and scanning EM. This combined multi-scale nano-imaging approach allowed the authors to observe structural damages in sperm caused by irradiation that are detrimental to its motility. This is an interesting study that provides potential links between irradiation and human reproductive health. It also showed the advantages of ptychography and cryo-SXT as alternative high-resolution ultrastructural imaging methods to view sub-cellular structures at their near-native state. However, several aspects in the manuscript may need to be discussed further, additional experiments are also needed to provide better support for the conclusions, some main concerns include:

1. Is 5 Gy irradiation a physiological condition that could be encountered under normal circumstances? Could spermatogenesis be re-established after longer time post-irradiation?
2. Although the authors claimed that the annulus is the main target of irradiation damage, it appeared that sperm suffered multiple damages to their structural integrity, including missing head and bent flagella, which all could contribute to the reduced motility, thus how could it be concluded that the annulus was the main target? Were mitochondria in the mid-piece also accumulate damages with time?
3. The sperm used for analyses were mainly from caudal epididymis, were sperm from other locations analyzed as well, including those within seminiferous tubule, since the irradiation was done on the whole body?
4. Some of the results need to be explained in more details, for example, in Figure 1, the parameters used to classify sperm abnormalities and how the percentages were calculated, e.g. annulus fracture. What are the colored rectangles in Fig. 1E?
5. How were sperm samples prepared for ptychography and cryo-SXT? More details on how the models were built using ptychography and cryo-SXT should be provided. Lines 153-171, not totally clear how the quantitative assessment of resolution of ptychography and cryo-SXT would help to visualize the ultrastructure of sperm.
6. Were the videos showing control samples or if irradiated samples could also be viewed in similar videos?

Reviewer #2 (Comments to the Authors (Required)):

The manuscript by Nuo Chen and colleagues reports on correlative imaging investigation of the high resolution structure related to the radiation-induced sperm dysfunction. The research has been performed with carefully planned and performed experimental work to chase the structure-function correlation in mouse sperm concerning the radiation induced damage, to discover the related high resolution structure evidence on the functional defect. In particular, multimodal approach to combine X-ray ptychography, cryogenic X-ray tomography, SEM and light microscopy images are overwhelming to deliver the main message with solid experimental observation. I can fully recommend the publication of the current work in the target journal, and leave comments for authors to consider for broad and clear delivery of the main work.

1. X-ray Ptychographic imaging results are presented but without description about what they are actually presenting. Are they intensity map (seemingly) or phase map? Here, I think the phase map from the ptychographic imaging can be more informative to quantitatively extract the structure information with clarity.
2. Cryogenic soft X-ray tomography images are impressive, but not much was described about how they collect, handle the data to obtain 3D images. Resolution estimated, and sample conditions etc need to be provided.
3. Some biological interpretation has to be made for the high resolution structural evidences discovered. For instance, annulus abnormalities in Fig. 3, what is the expected result for healthy sperm, reduced (scarred) diameter implies what?, etc.

Dear editor,

We sincerely thank the Editor and the Reviewers for their careful evaluation of our manuscript entitled “Annulus fracture underlies radiation-induced sperm dysfunction revealed by multimodal nano-imaging” (Manuscript ID: LSA-2026-03693-T). We are grateful for the constructive comments, which have helped us improve the clarity, rigor, and interpretation of the study. We have revised the manuscript accordingly and address each point below.

Reviewer 1

The authors reported in this manuscript analyses of sperm ultrastructure using multiple imaging methods including confocal microscopy, synchrotron radiation based soft X-ray ptychography, Cryo-SXT and scanning EM. This combined multi-scale nano-imaging approach allowed the authors to observe structural damages in sperm caused by irradiation that are detrimental to its motility. This is an interesting study that provides potential links between irradiation and human reproductive health. It also showed the advantages of ptychography and cryo-SXT as alternative high-resolution ultrastructural imaging methods to view sub-cellular structures at their near-native state. However, several aspects in the manuscript may need to be discussed further, additional experiments are also needed to provide better support for the conclusions, some main concerns include:

Comment 1: *Is 5 Gy irradiation a physiological condition that could be encountered under normal circumstances? Could spermatogenesis be re-established after longer time post-irradiation?*

Response: We thank the reviewer for raising this important point. In the present study, the choice of 5 Gy irradiation was based on the published work of Khan et al. (Khan S, Adhikari JS, Rizvi MA, et al. J Biomed Sci. 2015;22:61), in which 5 Gy was used as a sub-lethal whole-body γ -irradiation dose in male C57BL/6 mice to induce testicular injury. As stated in that study, “absorbed radiation dose in accidental sites and in planned exposure scenarios is less likely to be lethal,” and therefore investigation of sub-lethal doses such as 5 Gy is important for understanding radiation-induced reproductive injury. Based on this established model, we selected 5 Gy to generate reproducible and sufficiently severe sperm damage for multi-modal ultrastructural analysis.

At the same time, we agree that 5 Gy does not represent normal environmental or physiological radiation exposure. Rather, it should be considered a high-dose experimental/sub-lethal exposure condition, relevant mainly to accidental exposure scenarios, planned radiation exposure contexts, or mechanistic studies of radiation-induced reproductive toxicity. We have clarified this point in the revised manuscript.

Regarding the second question, available evidence indicates that spermatogenesis can be at least partially re-established after irradiation, depending on radiation dose, dose rate, and the post-irradiation interval. In particular, the study by Bae et al. (Bae MJ et al. Int J Mol Sci. 2021;22:12834) showed that radiation-induced impairment of mouse spermatogenesis is dynamic, and their analysis of testicular function at time points beyond 35 days after irradiation suggested that spermatogenic recovery may occur over time under certain conditions. These findings support the concept that the testis retains some regenerative capacity after radiation injury, likely depending on the survival of spermatogonial stem cells and the severity of the initial damage.

In our study, however, we focused primarily on the ultrastructural abnormalities of mature sperm revealed by multimodal imaging and their relationship to impaired motility at the selected post-

irradiation time point, rather than on long-term regeneration of spermatogenesis. Therefore, our current data do not directly determine whether spermatogenesis can be fully re-established after 5 Gy exposure. We agree that this is an important issue and have added this as a limitation and future direction in the revised Discussion.

Changes made in the manuscript:

- Page 12, Line 278: Added a statement in the Discussion explaining the rationale for selecting the 5 Gy radiation dose based on a previously established mouse model of radiation-induced testicular injury.
- Page 12, Lines 290~297: Added a statement in the Discussion noting that whether spermatogenesis can be re-established at longer post-irradiation time points remains to be further investigated.

Comment 2: *Although the authors claimed that the annulus is the main target of irradiation damage, it appeared that sperm suffered multiple damages to their structural integrity, including missing head and bent flagella, which all could contribute to the reduced motility, thus how could it be concluded that the annulus was the main target? Were mitochondria in the mid-piece also accumulate damages with time?*

Response: We thank the reviewer for this important comment. We agree that irradiation induces multiple sperm abnormalities, including head loss, hairpin/ring-like bending of the flagellum, and broader structural deterioration, all of which may contribute to impaired sperm function. We also acknowledge that our use of the term “main” in describing the annulus as the target of irradiation damage was overly definitive and therefore not sufficiently precise. We appreciate the reviewer for pointing this out.

Our conclusion regarding the annulus was based on the observation that annulus disruption was consistently associated with the hairpin/ring-like bending of the flagellum, suggesting that annulus injury is a key structural basis for this specific abnormal morphology. In this sense, we intended to indicate that the annulus is a prominent and functionally important site of damage, rather than the only damaged structure in irradiated sperm. We have therefore revised the corresponding wording throughout the manuscript to avoid overstatement.

Importantly, our fluorescence imaging and electron microscopy analyses further suggested that local membrane defects around the annulus region underlie the annulus disruption. We speculate that such membrane damage may destabilize the annulus and subsequently lead to the characteristic flagellar folding/bending phenotype. The same membrane fragility may also contribute to head-tail disconnection, which could explain the increased occurrence of missing-head sperm. However, because the exact breakage site between the sperm head and tail could not be directly resolved by our current multi-modal imaging workflow, this mechanism remains inferential and will require further validation by other approaches, such as additional high-resolution imaging or live/dynamic imaging strategies.

Regarding the mitochondria in the mid-piece, we carefully examined mitochondrial ultrastructure by transmission electron microscopy and did not observe obvious or significant mitochondrial structural alterations in our experimental setting. Therefore, based on our current data, mitochondrial damage does not appear to be the predominant ultrastructural feature in this model. To make this point clearer, we have added representative TEM images comparing mitochondria in control and irradiated sperm to the Supporting Information.

Changes made in the manuscript:

- Page 12, Lines 276-277: Replaced overly strong statements such as “the annulus is the main target” with more appropriate statements.
- Added representative TEM images of mitochondria in the mid-piece from control and irradiated sperm to the Supporting Information (Fig. S6), with the corresponding description incorporated into the Discussion (Page 12, Lines 287–290).

Figure 1. TEM images of mitochondria of control and irradiated sperm. Scale bar, 500 nm.

Comment 3: *The sperm used for analyses were mainly from caudal epididymis, were sperm from other locations analyzed as well, including those within seminiferous tubule, since the irradiation was done on the whole body?*

Response: We thank the reviewer for this helpful suggestion. In this study, we primarily analyzed sperm collected from the cauda epididymis because these spermatozoa are the fully matured cells that are most directly relevant to sperm motility and fertilizing function. Therefore, caudal epididymal sperm were the most appropriate specimens for our multi-scale structural and ultrastructural analyses.

At the same time, we agree that, because the irradiation was applied to the whole body, damage may also occur earlier during spermatogenesis within the seminiferous tubules. To address this point, we also performed histological analysis of the testes by H&E staining. The testicular sections showed a loss of elongated spermatids in the seminiferous tubules, indicating that irradiation impaired sperm development during spermatogenesis. These findings are also consistent with our histological observations in the epididymis. As described in the manuscript, H&E staining of the epididymis revealed more pronounced luminal sperm depletion in irradiated mice than in control mice,

particularly in the corpus epididymidis, where the epididymal duct lumen contained markedly fewer sperm (Fig. 1D). Together, these observations support the view that the abnormalities observed in mature sperm from the cauda epididymis likely originated, at least in part, from developmental defects arising in the testis and were further reflected by reduced sperm accumulation in the epididymal duct.

Therefore, although our high-resolution imaging analyses were mainly focused on mature sperm from the cauda epididymis, our additional testicular histological data support the conclusion that irradiation-induced sperm defects are already present during spermatogenic development in the seminiferous tubules.

Changes made in the manuscript:

- Added representative H&E-stained images of testicular tissue from control and irradiated mice to the Supporting Information (Fig. S7), with the corresponding description incorporated into the Discussion (Page 12, Lines 292–297).

Figure 2. Images of H&E staining of testicular tissue. Scale bar, 50 μ m.

Comment 4: *Some of the results need to be explained in more details, for example, in Figure 1, the parameters used to classify sperm abnormalities and how the percentages were calculated, e.g. annulus fracture. What are the colored rectangles in Fig. 1E?*

Response: We thank the reviewer for this helpful comment. We agree that the morphological classification criteria and quantification method should be more clearly described. Accordingly, we have revised the Figure 1 legend and the Materials and Methods section to explicitly define the morphological classification criteria and explain how the percentages were calculated.

Specifically, sperm were classified based on bright-field morphology as follows:

Morphologically normal sperm: sperm with an intact head, a continuous connection between the midpiece and principal piece, no obvious folding or bending of the flagellum into a nearly parallel configuration.

Annulus-fractured sperm: sperm showing an obvious discontinuity, narrowing, or break-like defect at the junction between the midpiece and principal piece (the annulus region).

Hairpin-/ring-like bent sperm: sperm in which the flagellum was folded at the annulus region to an

angle close to 180°, such that the midpiece and principal piece became nearly parallel.

Headless sperm: sperm flagella lacking an attached head.

For quantitative analysis, bright-field images of cauda epididymal sperm were collected from three mice per group. For each group, we analyzed at least 18 microscopic fields in total. The percentages of the normal and three abnormal categories (annulus-fractured, hairpin-/ring-like bent, and headless sperm) were calculated separately as the number of sperm in each category divided by the total number of sperm counted, multiplied by 100%. For example, percentage of annulus-fractured sperm = number of annulus-fractured sperm / total number of sperm counted × 100%

We have now added these definitions and calculation details to the revised manuscript to improve clarity and reproducibility.

Regarding Fig. 1E, the colored rectangles were used to indicate representative regions of headless sperm and hairpin-/ring-like bent sperm. We agree that this annotation was not sufficiently explained in the original figure legend, and we have now clarified this in the revised version.

Changes made in the manuscript:

- Page 14, Lines 363~374: Added a more detailed description of the classification criteria and quantification method in Materials and Methods.
- Page 4, Lines 100~101: Added the clarification of the colored rectangles in Fig. 1E.

Comment 5: *How were sperm samples prepared for ptychography and cryo-SXT? More details on how the models were built using ptychography and cryo-SXT should be provided. Lines 153-171, not totally clear how the quantitative assessment of resolution of ptychography and cryo-SXT would help to visualize the ultrastructure of sperm.*

Response: We thank the reviewer for this important comment. We have substantially revised the Methods and the relevant Results section to provide clearer details on:

(1) Sample preparation

The detailed protocol of sample preparation for ptychography and cryo-SXT was already presented in Materials and Methods in the part entitled by “Dehydrated and Cryogenic Sperm Sample Preparation”. Dehydrated sperm sample was prepared for ptychography experiment and cryogenic sample was prepared for cryo-SXT experiment.

(2) Reconstruction/model building

All projections were first aligned to the 0° reference projection using a cross-correlation method. After background removal, normalization was performed using the common-line method. CT reconstruction was then carried out with the real-space iterative reconstruction algorithm (RESIRE). After the initial reconstruction, all projections were further registered to the corresponding back-projection images via feature-point matching to achieve subpixel-level alignment correction. The final reconstruction was subsequently obtained using RESIRE.

(3) How the quantitative assessment of resolution of ptychography and cryo-SXT would help to visualize the ultrastructure of sperm

The resolution assessment was intended to clarify the structural information accessible by each imaging modality. Cryo-SXT provides nanometer-scale three-dimensional resolution, which allows visualization of the overall internal organization of sperm, including the mitochondrial sheath, axoneme, and fibrous sheath. Ptychography offers higher spatial resolution, enabling more detailed characterization of smaller structural features, particularly annulus morphology and local fracture-related changes at the midpiece–principal piece junction. Thus, the two techniques are

complementary in resolving sperm ultrastructure at different scales.

Changes made in the manuscript:

- Page 15, Lines 422~428: Added the 3D reconstruction method in Materials and Methods.

Comment 6: *Were the videos showing control samples or if irradiated samples could also be viewed in similar videos?*

Response: We thank the reviewer for this suggestion. The Movie S1 currently included in the manuscript corresponds to the 3D rendering of an **irradiated** mouse spermatozoon reconstructed from a 131-image tilt series. The Movie S2 currently included in the manuscript corresponds to the 3D reconstruction of a **control** spermatozoon generated from 38 projection angles using ptychographic computed tomography. We agree that providing a comparable video of an irradiated sample would improve the presentation and allow more direct visual comparison of structural alterations.

Reviewer 2

The manuscript by Nuo Chen and colleagues reports on correlative imaging investigation of the high-resolution structure related to the radiation-induced sperm dysfunction. The research has been performed with carefully planned and performed experimental work to chase the structure-function correlation in mouse sperm concerning the radiation induced damage, to discover the related high resolution structure evidence on the functional defect. In particular, multimodal approach to combine X-ray ptychography, cryogenic X-ray tomography, SEM and light microscopy images are overwhelming to deliver the main message with solid experimental observation. I can fully recommend the publication of the current work in the target journal, and leave comments for authors to consider for broad and clear delivery of the main work.

Comment 1: *X-ray Ptychographic imaging results are presented but without description about what they are actually presenting. Are they intensity map (seemingly) or phase map? Here, I think the phase map from the ptychographic imaging can be more informative to quantitatively extract the structure information with clarity.*

Response: We thank the reviewer for this important comment. We apologize for the ambiguity. The ptychographic images presented in the manuscript are phase-contrast reconstructions (phase maps), not intensity maps. We agree that this should have been explicitly stated because the phase signal is indeed the more informative quantity for quantitative structural analysis in weakly absorbing biological specimens.

In the revised manuscript, we now explicitly describe the ptychography results as 2D phase reconstructions generated from diffraction data using the mixed-state ePIE algorithm. We have also revised the corresponding figure legends to state “phase-contrast image” or “phase map” where appropriate.

Changes made in the manuscript:

- Revised the corresponding figure legends to state “phase-contrast image” or “phase map” where appropriate.

Figure 3. The intensity map and phase map of X-ray ptychographic imaging result.
Scale bar, 1 μm .

Comment 2: *Cryogenic soft X-ray tomography images are impressive, but not much was described about how they collect, handle the data to obtain 3D images. Resolution estimated, and sample conditions etc. need to be provided.*

Response: Regarding how to handle the data to obtain 3D images, all projections were first aligned to the 0° reference projection using a cross-correlation method. After background removal, normalization was performed using the common-line method. CT reconstruction was then carried out with the real-space iterative reconstruction algorithm (RESIRE). After the initial reconstruction, all projections were further registered to the corresponding back-projection images via feature-point matching to achieve subpixel-level alignment correction. The final reconstruction was subsequently obtained using RESIRE. The method of data collection, resolution estimation and sample conditions were made more clarified in the revised manuscript.

Changes made in the manuscript:

- Page 15, Lines 422~428: Added the 3D reconstruction method in Materials and Methods.
- The method of data collection (Page 14, Lines 400~421), resolution estimation (Supporting Information Fig. S3 & S4) and sample conditions (Page 14, Lines 384~399) were made more clarified.

Comment 3: *Some biological interpretation has to be made for the high-resolution structural evidences discovered. For instance, annulus abnormalities in Fig. 3, what is the expected result for healthy sperm, reduced (scarred) diameter implies what? etc.*

Response: We thank the reviewer for this valuable suggestion. We have strengthened the biological interpretation of the structural findings, especially in relation to Fig. 3.

In healthy sperm, the annulus forms a ring-like junctional structure at the boundary between the midpiece and principal piece and is expected to appear as an intact circumferential constriction/barrier maintaining local flagellar organization. In our measurements, the annulus diameter relative to the midpiece diameter is greater than 1 in controls, consistent with an intact annular ring architecture.

In irradiated sperm, the apparent reduction in annulus diameter relative to the midpiece reflects loss of annular continuity, i.e., a fractured or collapsed annulus with local discontinuity and exposure of underlying structures. Biologically, this implies weakening of the junctional scaffold and diffusion barrier, which may destabilize the membrane and fibrous sheath around the midpiece–principal

piece transition. This interpretation is reinforced by the reduction in SEPT12, localized membrane loss at the same region, and the progressive structural defects observed in adjacent flagellar compartments.

We have now incorporated this explanation into the Results.

Changes made in the manuscript:

- Page 7, Line 167~170: Added a more explicit biological interpretation of the annulus structural abnormalities, including the expected annulus morphology in healthy sperm and the potential implications of the reduced annulus diameter observed after irradiation.

Finally, we would like to thank the reviewers once again for the constructive comments and suggestions, which helped us significantly in improving the quality of the manuscript. We hope that we have addressed the reviewers' concerns properly, so that the manuscript can be accepted for publication in Life Science Alliance.

May 12, 2026

RE: Life Science Alliance Manuscript #LSA-2026-03693-TR

Prof. Huaidong Jiang
ShanghaiTech University
393 Middle Huaxia Road, Pudong New Area
Shanghai 201210
China

Dear Dr. Jiang,

Thank you for submitting your revised manuscript entitled "Annulus fracture underlies radiation-induced sperm dysfunction revealed by multimodal nano-imaging". Your manuscript was returned to Reviewer 2 who is satisfied with no further requests. In evaluating the revised manuscript, we feel that responses to two reviewer points provided in the rebuttal letter should be included in the manuscript text. First, the robust clarification provided to Reviewer 1 regarding the 5 Gy dose is absent in the revised text (5 Gy representing a high-dose exposure relevant mainly to accidental exposure scenarios, planned radiation exposure contexts, etc). Next, the structure of healthy sperm described in response to Reviewer 2 point 3 should be included in the manuscript text (the annulus forms a ring-like junction structure at the boundary between the midpiece and principal piece and is expected to appear as an intact circumferential constriction). We would be happy to publish your paper in Life Science Alliance pending these addition to the text and final revisions necessary to meet our formatting guidelines.

MANUSCRIPT ORGANIZATION AND FORMATTING:

To avoid unnecessary delays in the acceptance and publication of your paper, please read the following information carefully. Full guidelines are available on our Instructions for Authors page, <https://www.life-science-alliance.org/authors>

- Please remove the separate Supporting Information file containing supplementary figures. These should be only be uploaded as individual files.
- Please add an ORCID ID for corresponding author - you should have received instructions on how to do so.
- Please add a Summary Blurb/Alternate Abstract in our system.
- Please add the X and Bluesky handles of your host institute/organization, as well as your own, and/or one of the authors, in our system.
- Please rename "Competing Interest Statement" to "Conflict of interest" and move it along with the authors' contributions section after the Materials and methods section. Please refer to our manuscript guide: <https://www.life-science-alliance.org/manuscript-prep>
- It is recommended to exclude figures from the manuscript text and leave them uploaded separately.
- Please add your main, supplementary figure, and movie legends to the main manuscript text after the references section.
- We encourage you to revise the figure legends for figures S6 and S7 such that the figure panels are introduced in alphabetical order.
- Please add callouts for Figures 4C, F; S1A-B; S2A-E; S3A; S4A-D; S5A-F; S6A-D; S7A-D and movie S2 to your main manuscript text.
- Please include a "Data Availability" section after the Materials & Methods section. Please consult our guidelines at <https://www.life-science-alliance.org/manuscript-prep#format>
- Please remove Figure 7 and upload this figure as a Graphical Abstract.
- The Materials and Methods section is missing many important details. Please provide sufficient detail for a reader to reasonably reproduce your experimental procedures without referencing prior publications. Please include details on 3D Reconstruction methodology including software; Scanning electron microscopy including acquisition settings/parameters; Confocal fluorescence microscopy including staining protocol, light source, objectives, acquisition settings, and image processing.

We welcome submissions of potential cover images for the issue of LSA in which your work would appear. If you have high quality images associated with this work, please feel free to email these, with a caption, to the journal office.

LSA encourages authors to provide a 30-60 second video where the study is briefly explained. These videos will be appear embedded with the manuscript online at Life Science Alliance, and on social media to promote the published paper and authors (for examples, see <https://docs.google.com/document/d/1-UWCfbE4pGcDdcgzcmiuJl2XMBJnxKYeqRvLLrLSo8s/edit?usp=sharing>). Corresponding or first-authors are welcome to submit the video. Please submit only one video per manuscript. The video can be emailed to contact@life-science-alliance.org

FINAL FILES:

The following items are required for acceptance.

The license to publish form must be signed before your manuscript can be sent to production. A link to the license to publish form will be available to the corresponding author only. Please take a moment to check your funder requirements.

Thank you for your attention to these final processing requirements. Please revise and format the manuscript and upload materials as soon as you are able.

Thank you for this interesting contribution to the literature. We look forward to publishing your paper in Life Science Alliance.

Sincerely,

Reviewer #2 (Comments to the Authors (Required)):

The authors through this revised version have fully responded to all comments raised for the original submission. With this implementation, I can now fully support its publication in the current journal.

May 14, 2026

RE: Life Science Alliance Manuscript #LSA-2026-03693-TRR

Prof. Huaidong Jiang
ShanghaiTech University
393 Middle Huaxia Road, Pudong New Area
Shanghai 201210
China

Dear Dr. Jiang,

Thank you for submitting your Research Article entitled "Annulus fracture underlies radiation-induced sperm dysfunction revealed by multimodal nano-imaging". It is a pleasure to let you know that your manuscript is now accepted for publication in Life Science Alliance. Congratulations on this interesting work.

Your article will publish open access upon publication under a CC-BY license.

DISTRIBUTION OF MATERIALS:

Again, congratulations on a very nice paper. I hope you found the review process to be constructive and are pleased with how the manuscript was handled editorially. We look forward to future exciting submissions from your lab.

Sincerely,
